# Model Design and Applied Methodology in Geothermal Simulations in Very Low Enthalpy for Big Data Applications

Roberto Arranz-Revenga [1,*], María Pilar Dorrego de Luxán [2], Juan Herrera Herbert [1] and Luis Enrique García Cambronero [1]

1 Departamento de Ingeniería Geológica y Minera, Escuela Técnica Superior de Ingenieros de Minas y Energía, Universidad Politécnica de Madrid, 28003 Madrid, Spain; juan.herrera@upm.es (J.H.H.); luis.gcambronero@upm.es (L.E.G.C.)
2 Canal de Isabel II, Área de Infraestructura Informática, 28003 Madrid, Spain; mpdorrego@canal.madrid
* Correspondence: roberto.arranz@upm.es

**Abstract:** Low-enthalpy geothermal installations for heating, air conditioning, and domestic hot water are gaining traction due to efforts towards energy decarbonization. This article is part of a broader research project aimed at employing artificial intelligence and big data techniques to develop a predictive system for the thermal behavior of the ground in very low-enthalpy geothermal applications. In this initial article, a summarized process is outlined to generate large quantities of synthetic data through a ground simulation method. The proposed theoretical model allows simulation of the soil's thermal behavior using an electrical equivalent. The electrical circuit derived is loaded into a simulation program along with an input function representing the system's thermal load pattern. The simulator responds with another function that calculates the values of the ground over time. Some examples of value conversion and the utility of the input function system to encode thermal loads during simulation are demonstrated. It bears the limitation of invalidity in the presence of underground water currents. Model validation is pending, and once defined, a corresponding testing plan will be proposed for its validation.

**Keywords:** simulation; synthetic data; geothermal model; low enthalpy

## 1. Introduction

Currently, there is a growing demand for sustainable architecture, encompassing the utilization of eco-friendly materials and construction techniques, as well as the efficient use of energy. The use of geothermal heat pumps has increased to meet the demand for heat and domestic hot water (DHW). This approach aims to achieve increasingly higher levels of energy efficiency [1].

The use of very low-temperature geothermal energy is unevenly distributed worldwide. Even within the European Union, there is a significant difference among countries based on climate and per capita income. (This precludes any extrapolation of statistical data across regions).

A heat pump is defined as a thermal machine that uses a refrigerant gas in a closed thermodynamic cycle. In this way, it facilitates the flow of heat between the natural environment (air, water, or the earth) and a building or industrial facility. It also allows the reversal of the natural flow direction, from lower to higher temperature [2]. In the case of geothermal heat pumps, the subsoil is used as one of the two heat exchange sources. This takes advantage of the thermal stability and inertia that the ground offers from depths of approximately 15 to 20 m. By minimizing the thermal gradient between sources, the energy efficiency is maximized [3].

There are numerous implementations of geothermal HVAC (Heating, ventilating, air conditioned) systems, depending on thermal loads, HVAC circuits, DHW demand, and more. In these installations, heat exchange with the subsoil is typically accomplished by

means of vertical boreholes, typically ranging from 100 to 200 m in depth, with a minimum separation of 6 m between them. Thermal exchange is facilitated by heat exchangers installed within these boreholes, consisting of plastic pipes (commonly U-shaped) through which a brine solution circulates. The space between the borehole and the heat exchanger is sealed using a cement-based mortar. This mortar is enhanced with the addition of bentonite and silica, which reduces permeability and increases thermal conductivity [4].

Sizing the geothermal heat exchange field is a complex task, and in the process there are multiple uncertainties that are often non-trivial. Mathematical models are applied [5] to obtain approximations of the total length of the heat exchangers, considered the thermal loads applied to the building.

A thermal increase of 5 °C in the brine is considered during the design phase. Insufficient knowledge of geology and/or significant variations in the thermal load regime can lead to inaccurate forecasts. If this occurs after 50 years of operation [1], the average ground temperature can be significantly different from the initially predicted temperature.

To address these potential discrepancies, it is necessary to conduct a new study on the thermal behavior of the subsoil. The current investigation techniques, integrating artificial intelligence and big data, demand high-quality data. If geological data and thermal loads are to be used for the analysis of thermal exchange with the subsoil, the establishment of a data model is required to create high-quality datasets.

The process begins with an analytical approximation of thermal conduction in the ground surrounding a geothermal borehole. In a subsequent phase, a geometric model of potential geological structures involved in the system is developed. Finally, the model is converted into an electrical circuit that can be manipulated by any circuit simulation program. This approach allows the recreation of the system's operation and its long-term thermal response.

One of the main objectives of these simulations is to obtain the subsoil temperature distribution after a very long period of time. This means that the simulations are not aimed to size the heat exchanger, which is already defined and in operation. Different methods must be used compared to those used during the initial design. Although atmospheric conditions vary every year, it is necessary to consider the climate trend of recent years. Other factors, such us actual building occupancy, holiday periods, schedules, and so on, are also taken into account, which adds greater precision to the simulation and entirely eliminates unrealistic scenarios.

## 2. Materials and Methods

As a starting point, a homogeneous, thermally undisturbed subsoil is considered. The subsoil temperature remains stable throughout the year, starting from a depth of between 15 and 20 m. Heat flow is radial, and due to symmetry, the thermal diffusion law that defines the heat transfer problem can be expressed in polar coordinates, as shown in Equation (1).

The following constraints are applied:

- Water flows are not modeled, although there may be water-saturated materials;
- The average temperature ($T_1$) of the outer wall of the geothermal borehole is considered constant along its entire length;
- If the effect of the geothermal gradient of the ground is considered, it is treated as a boundary condition and not as an external heat source;
- The heat transfer mechanism between the borehole and the surrounding ground is solely thermal conduction;
- Only the heat flow between the ground and the borehole is considered. Its average value is the same along the entire borehole.

Taking these constraints into account, and supposing the borehole long enough and in a steady-state condition, we can simplify Equation (1) as shown in Equation (2), whose solution is given in Equation (3). In this context $T_1$ represents the temperature of the ground in contact with the geothermal borehole, with a radius of $r_1$. On the other hand, $T_2$ is the

temperature of the affected ground at $r_2$ from the borehole axis. In this case, $T_2$ coincides with the temperature of the thermally undisturbed ground[2], as can be observed in Figure 1.

$$\frac{1}{\alpha}\frac{\partial T}{\partial t} = \frac{1}{r}\frac{\partial}{\partial r}\left(r\frac{\partial T}{\partial r}\right) + \frac{1}{r^2}\frac{\partial^2 T}{\partial \theta^2} + \frac{\partial^2 T}{\partial z^2} \tag{1}$$

$$\frac{d^2 T}{dr^2} + \frac{1}{r}\frac{dT}{dr} = 0 \tag{2}$$

$$T = T_2 + \frac{(T_1 - T_2)}{Ln\left(\frac{r_1}{r_2}\right)}Ln\left(\frac{r}{r_2}\right) \quad [\text{K}] \tag{3}$$

Therefore, a cylindrical volume represents the ground affected by the borehole when the geothermal gradient is neglected. The axis of this cylinder corresponds to the geothermal borehole shown in Figure 1a. Adding the effect of the geothermal gradient in Equation (2) would only affect the terms that depend on $z$.

$$\frac{\partial^2 T}{\partial z^2} = 0 \tag{4}$$

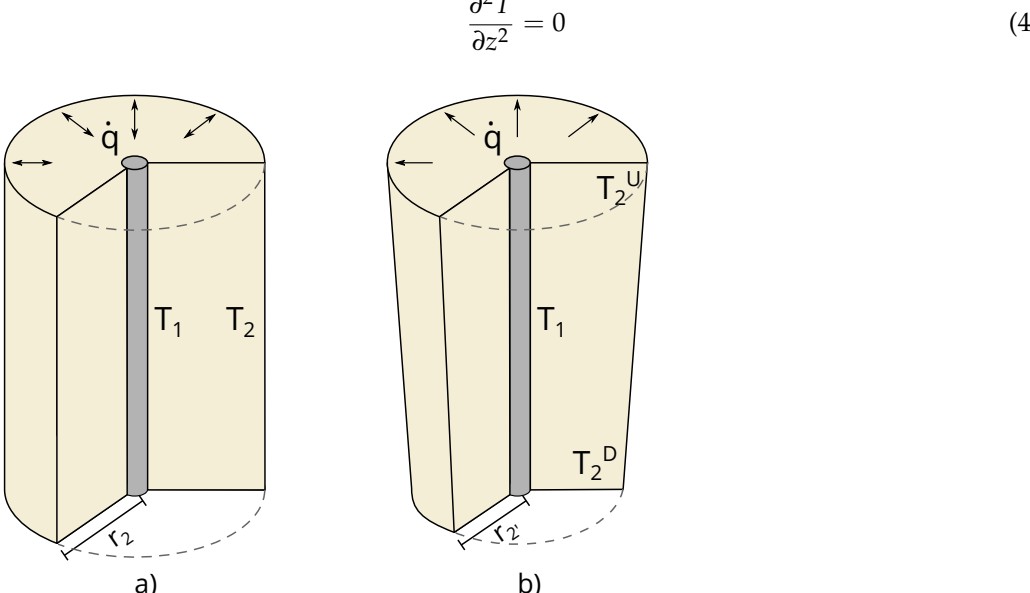

**Figure 1.** The shape of the homogeneous ground (light color) once it reaches a steady state. The vertical geothermal probe (dark color) with a radius of $r_1$ exchanges a heat flux $\dot{q}$ with the ground. (**a**) In this case, a zero geothermal gradient is considered. (**b**) If a geothermal gradient $\phi$ is applied, and there is a positive heat flux into the ground. Source: the authors.

The typical geothermal gradient in Earth is 0.03 °C/m [6], and in all cases, it remains constant, causing the value of Equation (4) to become zero. Therefore, Equation (2) remains valid. In this scenario, temperature $T_2$ increases with depth due to the geothermal gradient.

In the case where the geothermal heat exchanger operates solely in cooling mode, as shown in Figure 1b, only heat is transferred to the ground, and thermal equilibrium is reached faster at the bottom of the borehole. This means that $r'_2 < r_2$. In this case, the shape of the volume of ground affected by the geothermal borehole in a steady-state condition resembles that of a conical trunk tube with upper and lower radii of $r_2$ and $r'_2$, respectively.

The opposite occurs if a heat pump is used solely in heating mode.

The values of $r_2$ and $r'_2$ depend on the thermal properties of the materials and the heat flux. In a real-world scenario, steady state is rarely reached, different materials are usually encountered, and the thermal exchange regime often alternates between heat transfer and

heat absorption. Therefore, even though a direct analytical solution is not feasible, it serves as a foundation for developing a valid model.

In practice, steady state is not achieved due to the substantial thermal capacity of the soil and the magnitude of the exchanged energy. To determine a maximum radius of the affected ground and to work with simulations, an alternative approach must be considered. The maximum value for $r_2$ is typically set as half the distance between the two closest boreholes. At this point, the boreholes thermally interfere with each other.

### 2.1. Conversion of Heat into Electricity

Thermal conduction is governed by Fourier's law:

$$\dot{q} = -k\frac{dT}{dx} \quad [\text{W/m}^2] \tag{5}$$

Consider a wall with homogeneous thermal conductivity $k$ and thickness $L$. If the wall's surface is sufficiently large, the edges can be ignored. The electrical analogy of thermal conduction is illustrated in Figure 2, where Fourier's law (5) is replaced by Ohm's law (6) [7].

$$U = RI \quad [\text{V}] \tag{6}$$

In Figure 2, the wall is subjected to a temperature $T_1$ on the left-hand side surface, while there is a temperature of $T_2$ on the right side. According to the electrical analogy, temperature is now regarded as voltage, and the heat flow through the wall, from left to right, is considered an electric current. Here, $L/k$ represents the thermal resistance of the wall.

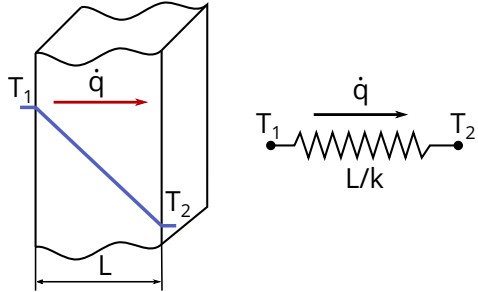

**Figure 2.** Equivalent electrical circuit of thermal conduction. Source: [8]. Modified.

In the event that the wall is thick, thermal capacity starts to become significant, and the phenomenon of thermal inertia emerges, a concept that has been applied in architecture since ancient times. Churches and other buildings with massive stone walls remain cool in summer and moderate in winter. In this scenario, an electrical capacitor is used as an analogy.

$$I = C\frac{dU}{dt} \quad [\text{A}] \tag{7}$$

Equation (7) defines the operation of the capacitor, with $C$ representing the capacitance measured in Farads [7]. The capacitor charges during the transient phase and discharges when the power supply changes direction or ceases. In the thermal analogy, it stores thermal energy when the ground is heated and releases the stored heat if cooling occurs. The opposite process occurs when the thermal conditions are reversed. In both cases, there is a slowing down of the thermal change due to external conditions.

### 2.2. Thermal Contact Resistance

When two layers of different conductive materials are in thermal contact, thermal resistance occurs at the interface. This is referred to as contact resistance. There is a significant temperature drop at the interface, which needs to be considered in heat transfer

calculations. The value of the contact resistance depends on the surface roughness and the pressure that holds the two surfaces together, among other factors [8].

In the example shown in Figure 3, you can observe how there is a temperature drop at the interface between two solid blocks in contact, which are laterally insulated from the exterior. The contact thermal resistance, $R_{AB}$, is measured in m²K/W, just like the other thermal resistances.

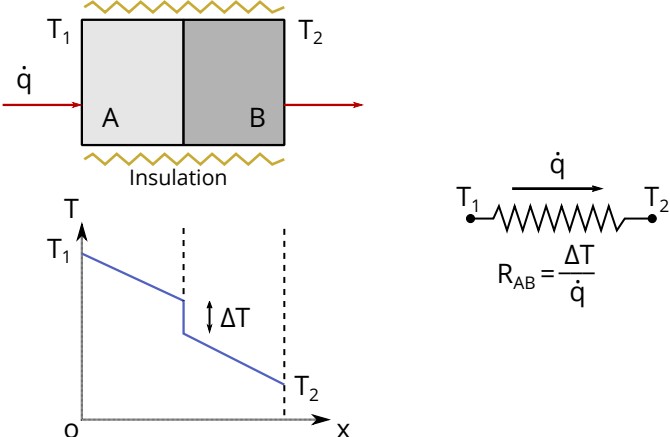

**Figure 3.** Equivalent electrical circuit of contact resistance. Source: [8]. Modified.

### 2.3. Single-Layer Ground

The simplest model that can be found is the one represented in Figure 4, which corresponds to the geothermal well and the surrounding ground.

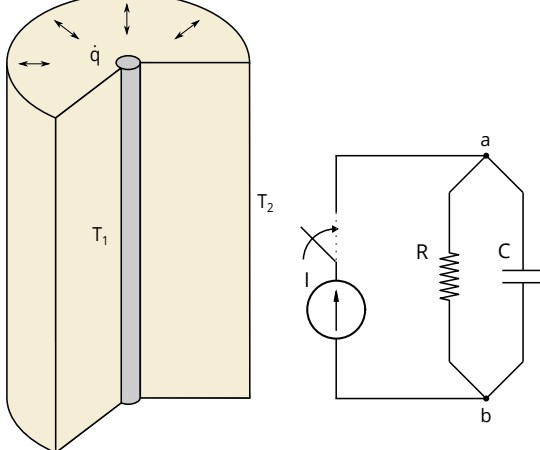

**Figure 4.** Electrical equivalent of a homogeneous layer of soil. Source: the authors.

The temperature field in the affected ground is studied, not that in the geothermal well itself. The geothermal gradient is not taken into consideration, and it is assumed that the geothermal well exchanges heat with the ground in a homogeneous and constant manner. A temperature difference exists between the cylindrical surfaces, with $T_1$ on the interior and $T_2$ on the exterior, of the affected ground.

Starting from an initial time t = 0, it is observed that $T_1$ equals $T_2$, at a depth ranging from 15 to 20 m. For a moment t > 0, a heat flow occurs between the geothermal well and the surrounding ground. A temperature difference emerges between the inner wall of the affected ground, $T_1$, and the external surface, $T_2$. This temperature difference is equivalent to the potential difference U between nodes a and b in the circuit shown in Figure 4.

As previously discussed, the heat flow direction is radial with respect to the axis of the affected ground cylinder. The flow direction depends on the temperature gradient.

Therefore, depending on the relative temperature of the geothermal well, the polarity of voltage $U$ and the direction of current flow $I$ will be determined.

In Figure 4 , the circuit operates in cooling mode and dissipates heat into the ground. During the ground recovery periods when the pump is not running, the switch is open, and the capacitor discharges, releasing heat into the system. It is assumed that the input or absorption of energy by the geothermal well is constant and zero when disconnected. Expressed in electrical terms, the system is subjected to a step current source. The circuit is completed with a resistance $R$, which corresponds to the thermal resistance of the ground, and a capacitor $C$, which represents the thermal capacity of the ground.

To characterize the equivalent electrical circuit, we begin with the equations defining electrical resistance (6) and electrical capacitor (7). The current I from the generator is divided at node (a) in Figure 1. The circuit is described by a first-order ordinary differential equation (ODE):

$$I = \frac{U}{R} + C\frac{dU}{dt} \quad [\text{A}] \tag{8}$$

In this case, the current is defined by a step function over time. The value of $I$ will be positive if the heat pump operates in cooling mode and negative if it operates in heating mode. On the other hand, the voltage $U$ between nodes (a) and (b) can be calculated from Equation (8).

### 2.4. Double-Layer Ground

Incorporating a second layer of material with distinct thermal properties makes the problem more complex. According to the new model, depicted in Figure 5, a thermal contact resistance emerges between the two layers due to irregularities on the surfaces [9]. Its electrical equivalent is denoted as resistance $R_{12}$.

Taking all these elements into account, the equivalent circuit of the new system is depicted in Figure 5, and the electrical circuit is described by Equations (9)–(11).

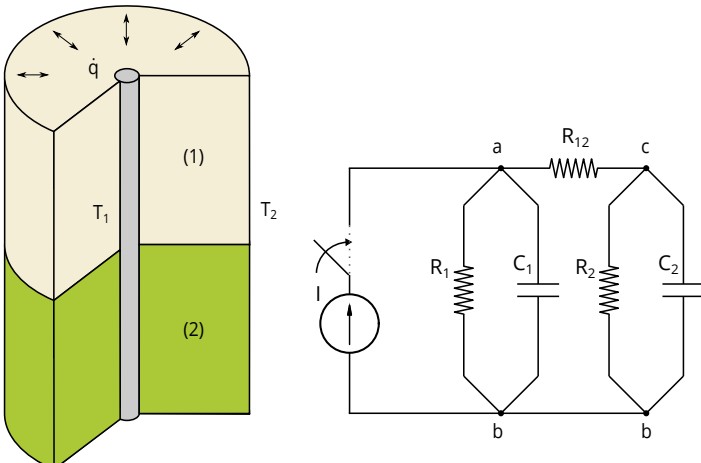

**Figure 5.** Electrical equivalent of two distinct homogeneous layers of soil. Source: the authors.

As in the previous case, there is an electrical current generator $I$, which corresponds to the geothermal well. An electrical resistance $R_i$ and a capacitor $C_i$ are present for each layer, and finally, the contact resistance between layers, $R_{12}$.

$$I = \frac{U_{ab}}{R_1} + C_1\frac{dU_{ab}}{dt} \quad [\text{A}] \tag{9}$$

$$I = \frac{U_{cb}}{R_2} + C_2\frac{dU_{cb}}{dt} + \frac{U_{ac}}{R_{12}} \quad [\text{A}] \tag{10}$$

$$U_{ab} = U_{ac} + U_{cb} \quad [V] \tag{11}$$

## 2.5. Enhanced Multilayer Ground

From the previous cases, a pattern in the equivalent electrical circuits can be observed. This enables the generalization of the circuit for n layers of different materials. An example of this generalized model can be seen in Figure 6, which illustrates a stratigraphic column composed of three geological materials (in three different colors). In this case, the column is divided into four layers of constant thickness, each experiencing the same temperature increment due to the geothermal gradient. Through a secondary subdivision (identified by letters), the boundaries between different materials and temperature ranges are adjusted. Thus, each letter indicates a type of material, and each number represents a temperature range.

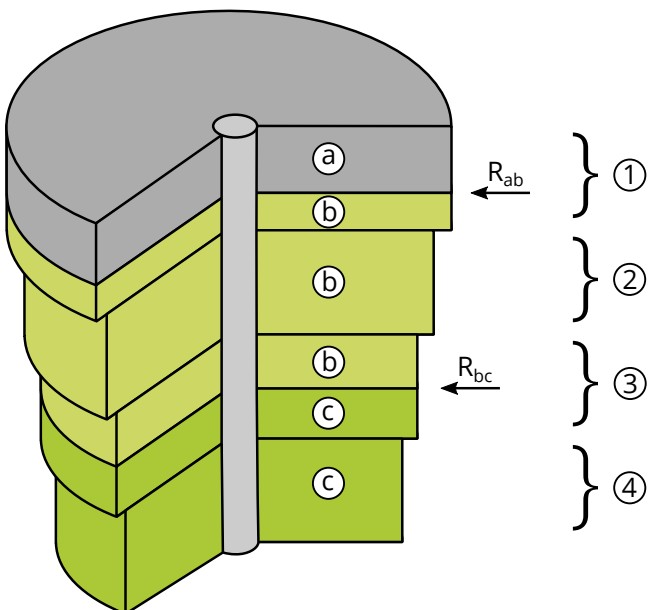

**Figure 6.** Geothermal well in thermally affected terrain divided into layers of constant thickness and variable sub-layers. In this example, three materials, a, b, and c, are depicted with two interfaces with contact thermal resistance R$_{ab}$ and R$_{bc}$. Source: the authors.

To enhance the model, the influence of the geothermal gradient is considered. Voltage generators have been added to simulate the increasing temperature with depth. One is added for each layer, and the voltage value increases progressively with depth in a constant manner. Nodes $y_1$ and $z$ represent the same point but are kept separate for clarity.

## 2.6. Thermal Load

In a geothermal system, the arrangement of thermal exchange boreholes is crucial and depends on the intended use of the installation. For instance, the distribution and spacing of boreholes for climate control differ from those used for heat storage [10]. Therefore, to make the data model useful, it is necessary to apply a set of constraints.

When using this model to simulate the thermal behavior of the ground, it is essential that the thermal loads be realistic and compliant with industrial regulations. These data are encoded in the form of an input function for the circuit. In summary, the following steps must be followed to obtain the thermal load:

- Establish the operating mode of the heat pump for the considered seasons following the corresponding industrial standard (e.g., UNE-EN 14511-1:2023 [11]);

- Once the operating hours are obtained according to the applicable industrial standard (e.g., UNE-EN ISO 52016-1:2017 [12]), distribute the assigned hours for a specific day according to the established schedule, considering whether it is a workday or a holiday and whether the building is occupied or not;
- With this information, create two files for hourly distribution and operating mode: one for heating and another for cooling.

One simple way to store data on the load regime of the heat pump is to use text files in CSV (Comma Separated Values) format. This file is imported into the simulation program and used as an input function. A possible structure of the file uses three types of data lines, with the first two as headers and the rest as actual data. Below are the parameters that make up this file:

Operating Mode (1 line) —The first line of the file, consisting of six values separated by commas, in the following order:

- Code—Text label identifying the operating mode according to the technical standard;
- $T_G$—Geothermal fluid temperature (°C);
- $T_E$—Building fluid temperature (°C);
- $P_T$—Thermal power (kW);
- $P_E$—Electrical power absorbed (kW);
- N—Number of operating days (0–366).

Occupancy Schedule (1 line)—The second line of the file. It stores the operating time range in minutes according to the type of day. It consists of nine values in the following order:

- W—Workday;
- $W_1$—Start minute (min);
- $W_2$—End minute (min);
- H—Weekend or holiday;
- $H_1$—Start minute (min);
- $H_2$—End minute (min);
- V—Vacation day;
- $V_1$—Start minute (min);
- $V_2$—End minute (min).

Operating Day (N lines) - From the third line of the file to (N+2). Each line consists of three values in the following order:

- D—Day of the year (1–366);
- $T_D$—Assigned schedule corresponding to weekdays, holidays, and vacations (D,H,V);
- M—Assigned operating time expressed in minutes (min).

## 3. Results

Since the article is a theoretical proposal, an example of applying the generalized circuit model and a practical calculation example of equivalent electrical values are presented. In both cases, the soil layer is one meter thick. To alter the thickness of the layer, the value of *L* needs to be changed.

### 3.1. Thermal to Electrical Data Transformation

As seen in Figure 2, the value of the electrical equivalent of the thermal resistance is $L/k$. In this case, the radius of the soil layer divided by *k* is used.

If a geothermal installation has several boreholes, the distance between them should range between 6 and 12 m, to avoid thermal interferences. Therefore, the value of the radius to calculate its thermal resistance is half the distance between the boreholes.

As an example, marl has been chosen as the material for the ground. Its average thermal conductivity value is 2.3 [13]. A separation of 12 m between geothermal wells

and a probe diameter of 160 mm has been chosen. Therefore, its equivalent electrical resistance is:

$$R_{\text{marl}} = \left(\frac{r_2 - r_1}{k}\right) = \left(\frac{6 - 0.08}{2.3}\right) = 2.574 \quad [\text{m}^2\text{K/W}] \tag{12}$$

For the equivalent electrical capacitance, the thermal capacity of the ground is used. Considering a volumetric heat capacity $C_v$ of the marl as 2.25 MJ/m$^3$K [13], under the same conditions as the previous example:

$$C_{\text{marl}} = VC_v\Delta T \quad [\text{MJ/m}] \tag{13}$$

$$\text{Where} \quad \begin{cases} V = \pi(r_2^2 - r_1^2)L \quad [\text{m}^3] \\ C_v = 2.25 \quad [\text{MJ/m}^3\text{K}] \\ \Delta T = 20 \quad [°\text{C}] \end{cases}$$

$$C_{\text{marl}} = 5088.474 \quad [\text{MJ/m}] \tag{14}$$

The thermal increase has been calculated, considering a steady subsurface temperature $T_2$ of 15 °C and an operation mode of the heat pump at 35 °C in the geothermal exchanger.

### 3.2. Geological-Electric Model

Taking into account what was discussed in the previous section, a geological model is proposed based on layers of different materials that are stacked on top of each other to form a stratigraphic column. Since each material has different thermal properties, when reaching a steady state, the radius of each disk is different. Materials with different thermal conductivity $k$ also lead to secondary heat flows between them. This process mainly occurs when the heat pump is idle, allowing the ground thermal recovery. The thermal influence zone of each borehole is greater than the typical distance between boreholes, as observed in Table 1. In practice, the steady-state condition is rarely achieved. To simplify, a cylinder with a maximum radius $r_2$ is considered equal to half of the minimum distance between boreholes.

**Table 1.** Comparison of the maximum influence radius of a geothermal well operating in a steady-state condition over different materials. Source of the K values: [6]. Source: the authors.

100 mm diameter borehole
Heat flow = 35 [W/m]
$T_1$ = 35 °C (borehole temperature)
$T_2$ = 15 °C (virgin ground temperature)

| Material | K [W/mK] | $r_2$ [m] |
|---|---|---|
| Ice | 1.20 | 5.95 |
| Basalt | 1.76 | 44.41 |
| Limestone | 2.21 | 223.42 |
| Claystone | 2.38 | 411.33 |
| Granite | 2.66 | 1124.07 |
| Marl | 2.69 | 1251.91 |
| Gneiss | 2.70 | 1297.67 |
| Marble | 2.80 | 1858.21 |
| Dolomite | 3.34 | 12,915.45 |
| Salt | 5.52 | 32,385,877.64 |
| Quartzite | 6.18 | 346,326,132.67 |

The truncated conical tube shape of each layer can be simplified as a cylindrical shape. To do this, the thickness of each layer will depend on the geothermal gradient and the resolution of the used thermal sensors. The layer thickness depends on the probe's sensitivity.

For example, with temperature sensors having a resolution of 0.5 °C and an average geothermal gradient of 0.03 °C/m, the thickness is $0.5/0.03 \approx 16.67$ m. Therefore, the thickness to be considered depends on the local geothermal gradient and the chosen temperature sensor's resolution. It is not necessary to have a built-in thermal sensor; only the resolution needs to be considered in the simulations. This determines the layer thickness as well as the temperature resolution obtained in the simulations.

When dividing the stratigraphic column into equal layers, the actual boundaries between materials do not coincide with these divisions. To address this issue, each layer is subdivided into sub-layers, using the boundaries between materials. The same approach is followed if a material spans more than one layer. An example of this type of division can be observed in Figure 6.

One way to utilize the layered model for simulations is to treat the problem as an electrical circuit. This is possible due to the thermo-electric equivalence that exists in these types of physical problems [14]. In this case, the elements to consider are resistors, capacitors, and a current source located at the geothermal borehole. At the interface of each material, a thermal contact resistance is applied ($R_{ab}$ and $R_{bc}$ in Figure 6). These resistances can be significant in some cases and must be taken into account in the simulation model.

In Figure 7, you can observe the electrical equivalent of the model shown in Figure 6. The circuit is divided into four zones that delineate the influence of the geothermal gradient. It is modeled using voltage generators multiplied by a number that increases in each layer. In the first layer, the multiplier is zero, and therefore, no voltage generator appears. These zones correspond to the division into layers of constant thickness observed in Figure 6. A second subdivision, identified by letters, models the different sub-layers of materials shown in Figure 6, which consist of a resistor and a capacitor connected in parallel.

The electrical circuit consists of a current source I (located on the left) controlled by the upper switch. The value of the current generator (positive or negative) is governed by a switching controller at the bottom.

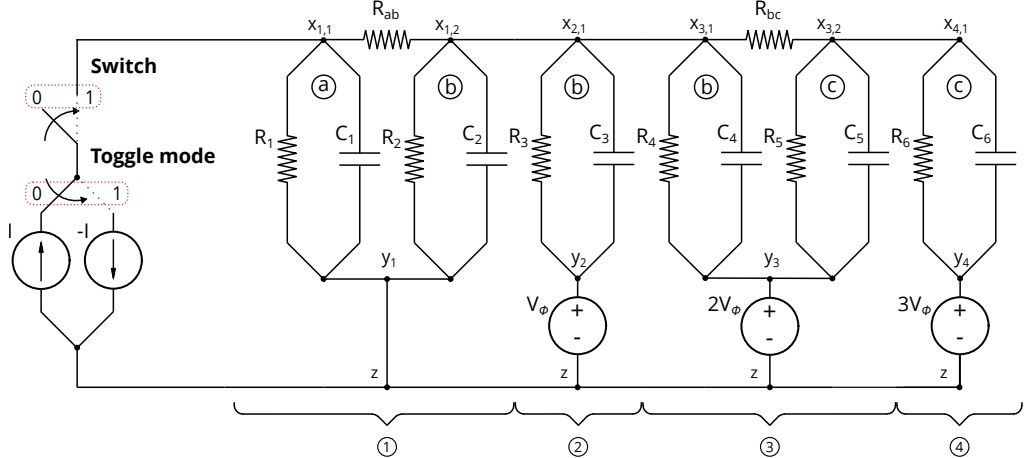

**Figure 7.** Electrical model corresponding to the geological-thermal model shown in Figure 6. Source: the authors.

In the circuit of Figure 7, voltage generators are visible to simulate the effect of the geothermal gradient, whose value increase proportionally to the depth. Each sub-layer has its own resistance and capacitance connected in parallel. At the material boundary, a contact resistance is inserted. This resistance is zero if the two adjacent sub-layers are made of the same material.

These types of circuits can easily be mimicked by means of specialized software. A non-periodic rectangular waveform is used as an input to the circuit. The amplitude indicates the value of I, and its (variable) period represents the geothermal pump's operation time and the periods of subsoil thermal recovery. Positive electric current values (+I) simulate heat dispersion in the ground, while negative values (−I) indicate heat absorbing, with

the geothermal heat pump acting as a heating system. After the simulation is completed, the values of voltage X*i*, *j* shown in Figure 7 are measured, and the arithmetic mean is calculated. By comparing this value with the mean at the beginning of the simulation, the variation in the average ground temperature is determined (reversing the temperature-electric voltage transformation). Another option is to take the values X*i*, *j*, providing a depth profile of temperatures.

In the production of synthetic data, it is possible to introduce random factors. These factors depend on the temporal distribution of usage and the climatic variation in the area of study. Therefore, it is of the utmost importance to pay attention to local characteristics. If the system is used with field geological data, the model can be adapted to the desired level of detail.

## 4. Discussion

Figure 8 shows the different phases of the research project, which culminate in the acquisition of a high-quality dataset. This article reflects the electrical model and the input function that will facilitate subsequent simulation. The validation of the electrical model is not covered in this outline or in this article.

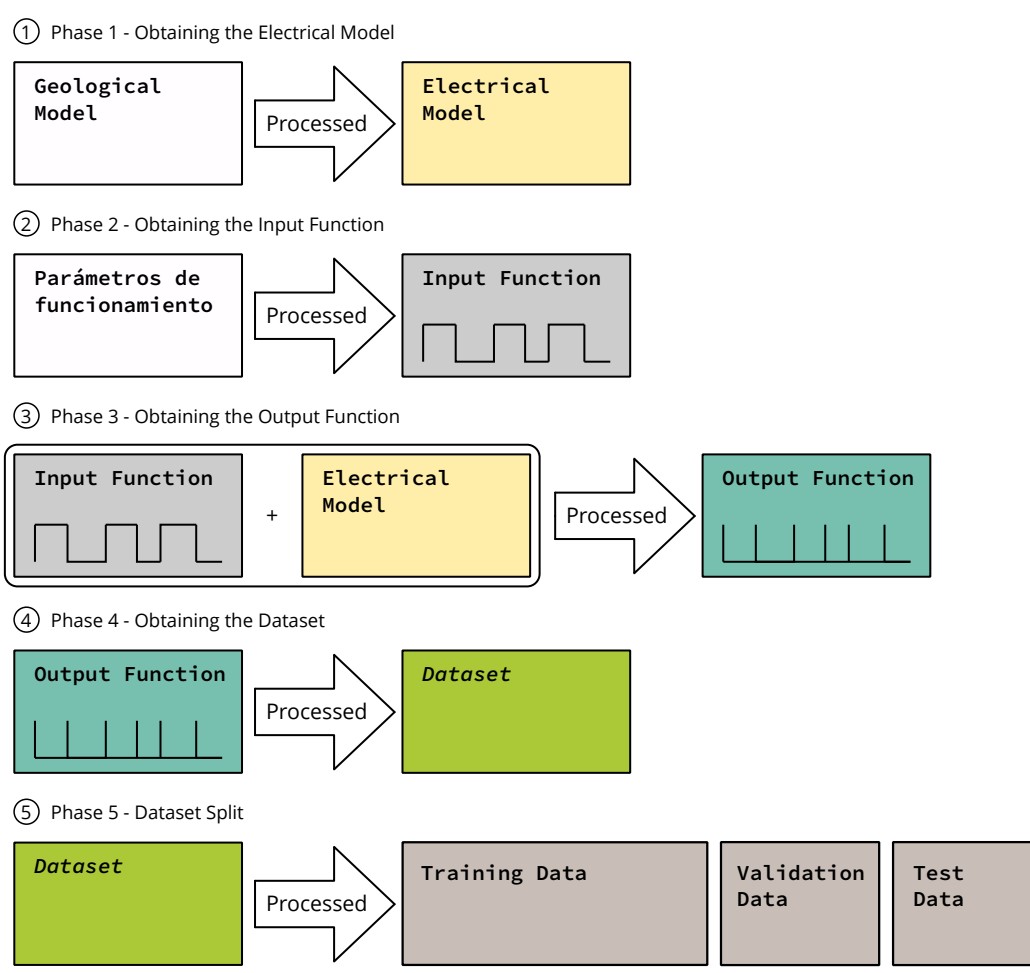

**Figure 8.** This diagram illustrates the various phases of the research project, which culminate in the acquisition of a high-quality dataset. Source: the authors.

The use of electrical circuits for simulation offers a significant range of commercial software options. Additionally, it enhances the scalability of the problem.

The generated synthetic data is useful on its own, beyond the initially proposed application. Daily schedule control enables the acquisition of specific data for studies in which building occupancy is crucial, such as energy optimization studies.

The use of a single input function in the simulator grants simulations of short periods or up to 50 years all at once.

One problem addressed by this model is the division of the heating calendar over multiple years. In the northern hemisphere, the heating season spans the end of one year and the beginning of the next. The electrical model simulation is controlled by a step input function. This function encompasses the time periods during which the thermal load is applied and its value. In this manner, the timeframe can be selected flexibly, akin to choosing frames in a movie.

The input function includes heating and cooling values, as well as periods of inactivity. However, it does not distinguish when it is used to produce domestic hot water (DHW). In these instances, it will be accounted for as heat or cooling, depending on the season.

## 5. Conclusions

A theoretical mathematical model has been proposed for shallow geothermal application. Although the ground may be water-saturated, it should not be intersected by underground currents. The model requires future validation based on data found in available literature or through independent experiments. The opportunity is given to other authors to use the proposed theoretical tool for their validation.

**Author Contributions:** Investigation, R.A.-R. and M.P.D.d.L.; writing—original draft preparation, R.A.-R.; writing—review and editing, R.A.-R. and J.H.H.; supervision, J.H.H. and L.E.G.C. All authors have read and agreed to the published version of the manuscript.

**Funding:** This research received no external funding.

**Institutional Review Board Statement:** Not applicable.

**Informed Consent Statement:** Not applicable.

**Data Availability Statement:** Data sharing in not applicable to this article.

**Conflicts of Interest:** The authors declare no conflict of interest.

## Abbreviations

The following abbreviations are used in this manuscript:

| | |
|---|---|
| DHW | Domestic hot water |
| HVAC | Heating, ventilating, air conditioned |
| CSV | Comma separated values |

## Notes

1. Reference period used in most of the regulations currently applied in the European Union.
2. Its numerical value approximates the average annual atmospheric temperature in the vicinity of the borehole.

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
