# Peer review of "Model Design and Applied Methodology in Geothermal Simulations in Very Low Enthalpy for Big Data Applications"

_data, 1980_

Round 1

Reviewer 1 Report

Comments and Suggestions for Authors

In this manuscript, the authors have presented the design of the data model and methodology applied in the simulation of very low-enthalpy geothermal systems for big data. The following points are to be addressed:

(a) The authors have presented a good model, and the methodology is very clear. But without any validation of the model, there is no meaning to the presented model. The authors need to validate the model either with earlier reported data or through a self-experiment.

(b) In an electrical model, how the authors will choose the values of resistance and capacitance with different layers should be eloborate.

(c) As the authors have mentioned, the problem addressed by this model is the division of the heating calendar over multiple years. Is there any specific solution proposed by the authors to resolve this?

Comments on the Quality of English Language

The paper is well written, however Minor editing of English language required.

Reviewer 2 Report

Comments and Suggestions for Authors

 The authors performed a good job in writing and analyzing the manuscript. As such, I am recommending it for publication.

Author Response

Thank you so much for taking the time to review this manuscript.

Reviewer 3 Report

Comments and Suggestions for Authors

This paper puts forward a new circuit idea to recalculate the service cycle of geothermal equipment, which is innovative in theory, but before further publication, there are still the following problems to be dealt with:

1. The abstract needs to be further improved, and it can't explain the contents of the manuscript well at present.

2. The documents listed in the references are not all listed in the manuscript, only 1-8 can be found.

3. There is a lack of references about the existing methods and standards, for example, UNE-EN 14511-1:2019.

4. The relevant models and methods put forward in the manuscript lack rationality verification.

5. Adding case analysis will make the manuscript more convincing.

Round 2

Reviewer 1 Report

Comments and Suggestions for Authors

Satisfactory